# Effect of Organic and Conventional Production on the Quality of Lemon "Fino 49"

Paola Sánchez-Bravo [1] , Luis Noguera-Artiaga [1],*, Juan Martínez-Tomé [2], Francisca Hernández [2] and Esther Sendra [1]

1   Research Group "Food Quality and Safety", Centro de Investigación e Innovación Agroalimentaria y Agroambiental, Miguel Hernández University of Elche, Carretera de Beniel km 3.2, 03312 Orihuela, Spain; paola.sanchezb@umh.es (P.S.-B.); esther.sendra@umh.es (E.S.)
2   Department of Plant Sciences and Microbiology, Centro de Investigación e Innovación Agroalimentaria y Agroambiental, Miguel Hernández University of Elche, Carretera de Beniel km 3.2, 03312 Orihuela, Spain; juan.martinez@umh.es (J.M.-T.); francisca.hernandez@umh.es (F.H.)
*   Correspondence: lnoguera@umh.es

**Abstract:** Since the end of the 20th century, organic foods have gained interest within the world population. The lemon is a fruit that is acquiring great prominence in the markets. Its use is based on its appreciated aroma and its content of bioactive compounds, but these attributes are greatly influenced by agronomic practices. To study the influence that organic farming has on the quality of "Fino 49" lemon variety in Spain, this citrus fruit has been characterized based on its morphological, functional, aromatic, and sensory properties. The results showed that conventional farming led to larger and elongated fruits (121.75 g of fruit weight and 58.35 and 79.66 mm of equatorial and polar diameter, respectively), with a higher lightness ($L^*$) in lemon skin (73.38) and higher content of organic acids (malic, citric, and succinic) and sugars (glucose and fructose). On the other hand, organic farming had a higher content of volatile compounds in lemon juice (2537 mg $L^{-1}$), especially limonene (increase of ~20%), which is related to the greater odor and flavor found in the organic samples by consumers (4.8 and 5.2, respectively). Furthermore, organic lemons had a better acceptance rate by consumers.

**Keywords:** citrus limon; consumer studies; organic acids; preference; volatile compounds

## 1. Introduction

Since 1970, organic foods have gained interest on a global scale [1]. Today, many farmers implement organic practices to attract consumers and increase their income. In turn, these agronomic practices promote crop rotation, biodiversity, etc., and they ban synthetic pesticides and fertilizers [1]. However, despite the fact that organic systems produce smaller volumes of food [2,3], some authors have shown that organic foods are more nutritious than conventional ones, presenting higher contents of vitamin C or total antioxidants, among other things [2,4–9]. In general, organic farming is considered to be more respectful of the environment than conventional farming, since it reduces the use of mineral fertilizers and pesticides, although the use of some non-synthetic pesticides, such as sulfur, copper sulphate, potassium permanganate, etc., are allowed [10]. Currently, they have increased the number of organic farms and therefore the extent of farmland. In the EU, the total area devoted to organic farming continues to increase, covering almost 13.8 million hectares of agricultural land in 2019 [11]. Likewise, the funds allocated to organic farming have also been increased [12].

Worldwide, the lemon (*Citrus limon* Burm.) is the third most cultivated citrus species, after orange and mandarin [13]. Since the end of the 20th century, the consumption of citrus fruits began to increase among the world population [14], reaching a production of more than 20 million tons in 2019 [15]. Spain is the second lemon producer country [16],

and the two most cultivated varieties in this country are "Fino" and "Verna", occupying almost the entire growing area destined for lemon. Lemons of the "Fino" variety are of high quality and are also used for international exports from October to February when prices are highest. "Verna" variety is harvested once the "Fino" harvest has finished and allows exports to be maintained during almost the entire period of the year [17].

Consumers are not only looking for a visually attractive fruit (size, color, firmness, etc.), but are also looking for internal quality (flavor, volatile compounds, functional compounds, etc.) [14,18]. Although citrus flavor and aroma are important factors in quality, consumers buy a lemon product based mainly on their perception of product quality and value for money [19].

In general, lemon is a fruit with an important content of bioactive compounds (vitamins, phenolic compounds, fiber, organic acids, mineral salts, etc.) responsible for beneficial health effects [20–22]. Several authors have reported its healthy properties on various types of cancer, cardiovascular diseases, obesity, cholesterol, etc. [20,23,24].

The flavor of a lemon depends on the concentration of organic acids and sugars and the relationship between them. Both are the main compounds in citric fruits. [25,26]. However, the main quality characteristic of citrus fruits is aroma [27]. Most of the volatile compounds of lemon are found in the skin of the fruit presenting 85–90% of the entire essential oil fraction of the lemon [28–30]. Citrus essential oils are used in the food industry as flavorings in beverages and alcoholic beverages, jams, jellies, sweets, ice cream and dairy products [31–33]. It is possible to determine direct relationships between the odor or flavor of a sample and the responsible volatile compounds in two ways: (i) comparative sensory analysis, using GC-MS to detect volatiles and find associations or (ii) using GC olfactometry ports -MS to detect and identify the responsible compounds [34].

On the other hand, the flavor, aroma, and functional quality of lemon depends on the agronomic practices, as use of rootstock, ripening, irrigation, etc., [18,19,25,31,35–37].

Therefore, the objective of this study was to determine if the type of cultivation (organic or conventional) affects the chemical, physical and volatile composition of "Fino 49" lemon.

## 2. Materials and Methods

### 2.1. Plant Material

The assay was carried out on lemons of the "Fino 49" variety grafted on *Citrus macrophylla*. "Fino 49" is a clone of the "Fino" variety and is the one that represents most young plantations (less than 10 years old) in the Mediterranean area. The Fino 49 variety is very early in its production, very productive and has a tendency to bear fruit in clusters. It has stumpy trees and shorter internodes than Fino but has larger thorns. The fruits are larger and heavier (170 g) and have a higher percentage of juice, with the possibility of harvesting in September–October. The rootstock used in the test is very vigorous, tolerant to Exocortis and very resistant to *Phytophotora*, but it is sensitive to *Tristeza* and *Xyloporosis*. Grafted with lemon trees, it is tolerant to all serious virosis, except *Xyloporosis*. It presents a very good affinity with all varieties. It is resistant to limestone and salinity and is sensitive to root suffocation and very sensitive to frost. The characteristics that stand out the most are the rapid entry into production, high productivity, and early ripening, with average fruit quality. Organic growing was carried out on the "Lo Vera" orchard (38°03′24.6″ N, 0°44′25.0″ W) with an area of 10.28 ha and conventional cultivation was realized on the "Lo Lorente" orchard (38°02′21.3″ N, 0°44′51.3″ W) with an area of 9.11 ha. Both orchards were located in Alicante (Spain), at a distance of ~2 km, and the tree spacing was 6.5 × 4 m. The trees are 10 years old (both farms were planted in 2011). The climate is characterized by mild winters and hot summers, temperatures ranging between 33 and 10 °C, and light rains concentrated in spring and autumn.

The organic farming orchard has a clay-textured soil (sand 29.55%, silt 30.00% and clay 40.45%), with an organic matter content of 1.40%. The nitrate content is 62.20 mg kg$^{-1}$, the Olsen assimilable phosphorus content is 15.8 mg kg$^{-1}$ and potassium in the aqueous extract is 1.07 meq L$^{-1}$. The cation exchange capacity is 14.70 meq 100 g$^{-1}$. The C/N ratio

is 7.84 and the field capacity is 27.60 (% dry soil). The conductivity of the 1:2 aqueous extract at 25 °C is 1.21 dS m$^{-1}$.

The conventional orchard has a loam-clay-loam texture soil (sand 12.05%, silt 52.50% and clay 35.45%), with an organic matter content of <0.65 %. The nitrate content is 88.10 mg kg$^{-1}$, the Olsen assimilable phosphorus content is 5.37 mg kg$^{-1}$ and potassium in the aqueous extract is 0.39 meq L$^{-1}$. The cation exchange capacity is 10.40 meq 100 g$^{-1}$. The C/N ratio is 5.93 and the field capacity is 28.40 (% dry soil). The conductivity of the 1:2 aqueous extract at 25 °C is 0.31 dS m$^{-1}$.

As for the water used, for the orchard destined for organic farming, it had an electrical conductivity of 1.16 dS m$^{-1}$ and a pH of 8.4. The cation content expressed in mmol L$^{-1}$ was 1.74 for calcium; potassium 0.21; magnesium 1.87 and sodium 4.31. The anion content expressed in mmol L$^{-1}$ was <0.02 for nitrates; phosphates <0.01; sulfates 2.44; bicarbonates 3.00 and chlorides 4.37.

The water of the conventional orchard had an electrical conductivity of 1.32 dS m$^{-1}$ and a pH of 8.1. The cation content expressed in mmol L$^{-1}$ was 2.12 for calcium; potassium 0.27; magnesium 1.91 and sodium 4.83. The anion content expressed in mmol L$^{-1}$ was <0.02 for nitrates; phosphates 0.05; sulfates 2.98; bicarbonates 3.39 and chlorides 4.54.

Treatments to control pests and diseases at the organic and conventional orchards are in Table 1.

**Table 1.** Treatments to control pests and diseases at experimental orchards.

| Date | Pests & Diseases | Treatments |
|---|---|---|
| Organic | | |
| 12-April | *Prays citri* | *Bacillus thuringiensis* Var. Kurstaki (PB-54 Strain) 32 × 106 |
| 05-May | *Prays citri* | *Bacillus thuringiensis* Var. Kurstaki (PB-54 Strain) 32 × 106 |
| 01-October | *Phytophthora citrophthora* | 35% cupric hydroxide |
| Conventional | | |
| 15-February | *Aonidiella aurantii* & *Tetranychus urticae* | 83% paraffin oil & 25.87% Hexitiazox |
| 12-April | *Prays citri* | *Bacillus thuringiensis* Var. Kurstaki (PB-54 Strain) 32 × 106 |
| 05-May | *Prays citri* | *Bacillus thuringiensis* Var. Kurstaki (PB-54 Strain) 32 × 106 |
| 24-May | *Aonidiella aurantii* & *Tetranychus urticae* | 83% paraffin oil, 25.87% Hexitiazox, Abamectin 1.8% and Spirotetramat 10%. |
| 01-October | *Phytophthora citrophthora* | 35% cupric hydroxide |

Regarding fertilization, the fertilization formula in the two farms was as follows: 257 kg ha$^{-1}$ of N, 47 kg ha$^{-1}$ of P$_2$O$_5$ and 160 kg ha$^{-1}$ of K$_2$O. The fertilizers used in the organic orchard were:

- Solublack H 87 (70% humic acids and 15% fulvic acids).
- Brotolim eco N-AA (6% N + 12% AA + 30% MO).
- Ourpizca (CaO 6%).
- Solured Mn-2.5.
- Solured Zn-2.6.
- Unicquel (Iron chelate 6%).
- Haifa sop bio (K$_2$O 53% + SO$_3$ 40%).
- Agrosol fluid 2-4-6 + AA.
- Copper shuttle (Cu 6.13%).

The fertilizers used in the conventional production were:

- Brotolim primavera + Ca + Mg (10-2-5 + 3.5 CaO + 1.5 MgO).
- Unicquel (Iron chelate 6%).
- Novatec fluid engorde (7,5-2,5-6,8).
- Brotolim Ca AC (CaO 8%).

- Brotolim engorde + Ca (8-2-6 + 2,5 CaO).
- Lumbusol (organic matter 30%).

The sampling of the fruits took place in the second half of October 2021. For each orchard (organic and conventional) 100 lemons from 10 trees (10 fruits per tree) were hand-harvested (from the middle part and the entire perimeter area of each tree) at physiological maturity, and immediately transported to the laboratory. Once in the laboratory, a selection of the 75 most homogeneous fruits (size, shape, and color) was made and 50 of these were separated for the physical-chemical, organic acids, sugars, and volatile compounds determinations, and the other 25 fruits were used for the sensory analysis.

### 2.2. Physical and Chemical Analysis

A toal of fifty lemon fruits from each grow system were selected to determine color, pH, size, total soluble solids (TSS), total titratable acidy, and weight. For the determination of the lemons' weight (FW), a Mettler balance model AG204 scale (Mettler Toledo, Barcelona, Spain) was used. A digital caliper (model 500-197-20, 150 mm; Mitutoyo Corp., Aurora, IL, USA) was used for the determination of the size of each fruit: peel thickness (PT), polar diameter (PD), and equatorial diameter (ED). The lemon's color was measured in peel and juice. Peel color was measured in CIE *L*a*b** coordinates (*L**, *a**, *b**) along the equatorial axis of each fruit (the same 50 used for the measurement of size and weight) using a calibrated Minolta CRC-200 (Minolta Camera Co. Osaka, Japan) spectrophotometer, with illuminated D65 and an observer of $10°$. The results were expressed according to the CIE *L*a*b** system (CIE, 1931). The mean values for coordinates *L** (lightness), *a** (red-green), *b** (blue-yellow), *C** (Chroma) and h (tone) and with them, the corresponding citrus color index (CCI = 1000 $a*/L*b*$) using the equation proposed by Jiménez-Cuesta, et al. [38] were calculated. Color parameters in CIE *L*a*b** were also measured on the lemon juice placed in a liquid accessory CR-A70. Total soluble solids (TSS) content ($°$Brix) was measured using a digital Atago refractometer (model N-20; Atago, Bellevue, WA, USA) at 20 $°$C. Total titratable acidy and pH were determined by acid–base potentiometer (877 Titrino plus, Metrohm ion analyses CH9101, Herisau, Switzerland), using 0.1 mol $L^{-1}$ NaOH up to a pH of 8.1; results were expressed as g of citric acid $L^{-1}$. Measurements were determined in triplicate.

### 2.3. Organic Acids and Sugars

Organic acids and sugars were identified and quantified according to Hernández, et al. [39] with some modifications. The lemon juices were prepared by hand-squeezing in a commercial juicer. Then, the juices were centrifuged at 15,000× *g* for 20 min (Sigma 3–18 K; Sigma, Osterode am Harz, Germany), filtered through a 0.45 μm Millipore filter, and injected (10 μL) into a high-performance liquid chromatograph (HPLC) system (HP Series 1100, Hewlett-Packard, Wilmington DE, USA). Organic acids were separated on a Supelcogel C-610H column (30 cm × 7.8 mm) fitted with a Supelcoguard column (5 cm × 4.6 mm) (Supelco, Inc., Bellefonte, PA, USA) and a diode array detector (210 nm) (HPLC, Waldbronn, Germany) was used for the detection. For the determination of sugars, the same HPLC system and conditions were used, but detection was performed with a refractive index detector (HPLC, Waldbronn, Germany). Standard curves of pure organic acids and sugars were used for quantification. Concentrations of both organic acids and sugars were expressed as g 100 mL$^{-1}$ of juice. Determinations of sugars and organic acids were conducted in triplicate.

### 2.4. Volatile Compounds

The volatile compounds of the lemon juice and essential oil obtained from the peel were analyzed. The determination of volatile compounds of the essential oil was carried out following the method described by Aguilar-Hernández, Sánchez-Bravo, Hernández, Carbonell-Barrachina, Pastor-Pérez and Legua [18]. The extraction of the volatile compounds of the samples of lemon juice was carried out using the headspace solid-phase micro-

extraction (HS-SPME) method. For the extraction, a SPME 50/30 mm DVB/CAR/PDMS (Divinylbenzene/Carboxen/Polydimethylsiloxane) fiber (Supelco) was used, whose exposure time was 40 min at a temperature of 45 °C and with constant agitation (500 rpm) using a magnetic stirrer (IKA C-MAG HS 4, IKA-Werke GmbH & Co. KG, Staufen, Germany).

A chromatograph Shimadzu GC2030 (Shimadzu Scientific Instruments, Inc., Columbia, MD, USA) with an SLB-5 MS column (Teknokroma, Barcelona, Spain) was used to separate the volatile compounds. The column had a length of 30 m, an internal diameter of 0.25 mm, and a film thickness of 0.25 μm. For the identification of compounds, the chromatograph was coupled with a Shimadzu TQ8040 NX mass spectrometer detector. The parameters of the mass spectrometer were: (i) mass range 45–400 m/z, (ii) scan speed 5000 amu/s, (iii) event time of 0.200 s, and (iv) electronic impact of 70 eV. Helium was used as a carrier gas, with a split ratio of 1:10, a purge flow of 6 mL min$^{-1}$, and a total column flow of 0.6 mL min$^{-1}$. The temperature of the detector was 300 °C, and the temperature of the injector was 230 °C. The total desorption time of the sample in the injection port was 3 min.

The oven program was the following:

- Initial temperature of 80 °C.
- Ramp of 3 °C min$^{-1}$ from up to 170 °C.
- Increase of 25 °C min$^{-1}$ up to 300 °C, and maintained for 50 s.

A commercial alkane standard mixture (Sigma-Aldrich, Steinheim, Germany) was used to calculate the retention indexes (Kovat's index). NIST 17 Mass Spectral and Retention Index libraries were used for the identification of compounds. Only compounds with spectral similarity >90% and with a deviation of less than 10 units of linear retention similarity were considered as correct hits. The analysis of the volatile compounds of the lemon samples was run in triplicate.

### 2.5. Sensory Analysis

A consumer's study was carried out with 100 participants. The juice samples were diluted (1 mL in 25 mL of water) for analysis. All samples were served at a temperature of 8 °C and encoded using three aleatory digit codes. Between each of the samples, osmotic water, and crackers (unsalted) were used to clean the palate. The test room had a temperature of 22 °C and a combination of natural and non-natural (fluorescent) light. Consumers were asked about color, lemon odor, lemon flavor, sourness, bitterness, and aftertaste. In addition, consumers indicated their "global" satisfaction and purchase intention for the samples under study. Demographic data were also collected. A 9-point scale was used, where 9 = like extremely and 1 = dislike extremely.

### 2.6. Statistical Analysis

A one-way analysis of variance (ANOVA) and Tukey's multiple range test were performed for the analysis of the results. XLSTAT software (2016.02.27444 version, Addinsoft: New York, NY, USA) was used for the statistical treatment of the data. The confidence interval was 95% and the significant difference was defined as $p < 0.05$. All of the determinations were run in triplicate.

## 3. Results and Discussion

### 3.1. Physical and Chemical Analysis

Growing lemon in a conventional way has been shown to obtain fruits with greater weight (121.75 g), greater fruit equatorial diameter (58.35 mm), and fruit polar diameter (79.66 mm) than fruits grown organically (Table 2). On the other hand, it was organic lemons that showed a better relationship between equatorial and polar diameter (0.78) and a lower skin thickness (5.14 mm). These values were slightly lower than those reported by Aguilar-Hernández, Núñez-Gómez, Forner-Giner, Hernández, Pastor-Pérez and Legua [17].

**Table 2.** Physical parameters of lemon "Fino 49" cultivated under organic and conventional farming.

|  | FW (g) | ED (mm) | PD (mm) | ED/PD | PT (mm) |
|---|---|---|---|---|---|
| Conventional | 121.75 a | 58.35 a | 79.66 a | 0.74 b | 5.53 a |
| Organic | 106.17 b | 56.32 b | 72.34 b | 0.78 a | 5.14 b |
| ANOVA | *** | ** | *** | ** | * |

*, ** and ***: significant at $p < 0.05$, 0.01 and 0.001, respectively. Values followed by different letters, within the same column, were significantly different ($p < 0.05$). FW (fruit weight), ED (fruit equatorial diameter), PD (fruit polar diameter), PT (peel thickness).

Overall, color is considered a determining factor by the consumer for their purchase. This color change occurs due to the degradation of chlorophylls present in the skin and the synthesis of carotenoids, which are the ones that provide the yellow color to citrus fruits [40]. The color obtained for the evaluated lemon skin samples was like that obtained by Di Matteo, Di Rauso Simeone, Cirillo, Rao and Di Vaio [13] in Italian cultivars (Table 3). In general, there were no significant differences between the samples (skin and juice), except for the lightness parameter (*L**), in which conventional lemons presented the highest values (73.38). However, it should be noted that differences of less than three units are imperceptible to the human eye [41], and therefore, it is not a notable difference. The high values in the parameter *b** indicate a yellowish color in the peel of the evaluated lemons [42], both conventional and organic. Likewise, the negative values in the *a** parameter in the juice samples are representative of pale juices with slight greenish tones [17].

**Table 3.** Color analysis of lemon skin and lemon juice under organic and conventional farming.

|  | *L** | *a** | *b** | *C** | h | CCI |
|---|---|---|---|---|---|---|
| *Lemon Skin* |  |  |  |  |  |  |
| Conventional | 73.38 a | −4.42 | 58.49 | 59.06 | 83.37 | −1.26 |
| Organic | 70.92 b | −5.88 | 56.73 | 57.55 | 82.21 | −1.74 |
| ANOVA | * | NS | NS | NS | NS | NS |
| *Lemon Juice* |  |  |  |  |  |  |
| Conventional | 30.77 | −1.31 | −0.23 | 1.34 | 10.72 | 322.31 |
| Organic | 33.44 | −1.48 | 0.17 | 1.51 | 10.63 | 15.27 |
| ANOVA | NS | NS | NS | NS | NS | NS |

NS: not significant at $p > 0.05$; *: significant at $p < 0.05$. Values followed by different letters, within the same column, were significantly different ($p < 0.05$). *L** (lightness), *a** (green-red coordinate) and *b** (blue-yellow coordinate), *C** (Chroma) and h (tone), CCI (Citrus Color Index = 1000 $a*/L*b*$)).

Some authors have compared the color of conventional and organic fruits. It has been found that the use of fertilization (nitrogen) could affect the color of these fruits, due to the increase of nitrogenous substances and the lower content of carbohydrates in fruits (apples and grapes), or due to decreased activity of enzymes that modulate pigment synthesis (grapevines and mangoes) [43].

### 3.2. Total Soluble Solids (TSS), pH, Total Titratable Acidity, Organic Acids and Sugars

The relationship between the concentration of sugars and organic acids is the main cause of the lemon flavor [14]. Total soluble solids (TSS), pH and total titratable acidity (TA) of the lemons grown by conventional and organic farming were analyzed (Table 4). In general, the flavor of citrus fruits is highly influenced by the level of TSS and organic acids [44]. In addition, TA is used as an indicator of the quality of juices [45]. Organic lemons had a lower TSS (8.77 °Brix) and total titratable acidity (57.27 g citric acid $L^{-1}$) concentration than lemons grown in a conventional way (9.77 °Brix and 64.39 g citric acid $L^{-1}$). However, both samples presented values higher than those defined by García-Sánchez, et al. [46] and Marín, et al. [47]. There were no significant differences regarding pH. Similar results on a lower total acidity content in samples from organic farming were also found in oranges [48], apples [49], and strawberries [50].

**Table 4.** Total soluble solids (°Brix), pH, total titratable acidity (TA; g citric acid $L^{-1}$), organic acids (g $L^{-1}$) and sugars (g $L^{-1}$) of lemon "Fino 49" cultivated under organic and conventional farming.

|  | °Brix | pH | TA | Malic | Citric | Succinic | Glucose | Fructose |
|---|---|---|---|---|---|---|---|---|
| Conventional | 9.77 a | 2.94 | 64.39 a | 0.48 a | 6.04 a | 0.65 a | 10.67 a | 5.50 a |
| Organic | 8.77 b | 2.91 | 57.27 b | 0.27 b | 5.45 b | 0.56 b | 9.49 b | 4.93 b |
| ANOVA | ** | NS | ** | *** | *** | ** | *** | ** |

NS: not significant at $p > 0.05$, ** and ***: significant at $p < 0.01$ and 0.001, respectively. Values followed by different letters, within the same column, were significantly different ($p < 0.05$).

Two sugars (glucose and fructose) and three organic acids (malic, citric and succinic) were identified and quantified in the lemon samples under study (Table 4). The samples grown under organic conditions presented a lower concentration in all the sugars (9.49 g $L^{-1}$ of glucose and 4.93 g $L^{-1}$ of fructose for the organic sample, and 10.67 g $L^{-1}$ of glucose and 5.50 g $L^{-1}$ of fructose for the conventional sample) and organic acids (0.27 g $L^{-1}$ of malic acid, 5.45 g $L^{-1}$ of citric acid and 0.56 g $L^{-1}$ of succinic acid for the organic sample, and 0.48 g $L^{-1}$ of malic acid, 6.04 g $L^{-1}$ of citric acid and 0.65 g $L^{-1}$ of succinic acid for the conventional sample) were detected. Similar values for sugars were reported by Serna-Escolano, Martínez-Romero, Giménez, Serrano, García-Martínez, Valero, Valverde and Zapata [14], although these authors also found sucrose in the composition of sugars in lemon juice. These results agree with those obtained in the TSS and TA analyses. The same result, in terms of a higher content of citric acid in the fruits obtained through traditional cultivation, was obtained by Uckoo, et al. [51] in lemons (*Citrus meyeri* Tan).

### 3.3. Volatile Compounds

A few years ago, the essential oil of lemons was studied solely due to its flavor and aroma, seeking to supplement other foods with it. This situation is changing and essential oils are gaining interest [20].

A total of thirty-two volatile compounds were identified in the volatile profile of lemon juice under study (Table 5) and thirty-four compounds were identified in the lemon skin oils samples (Table 6). In the lemon juice samples, the three most abundant compounds were limonene (1310 and 1086 mg $L^{-1}$, organic and conventional, respectively), γ-terpinene (252.85 and 188.49 mg $L^{-1}$, organic and conventional, respectively) and α-terpineol (198.45 and 143.26 mg $L^{-1}$, organic and conventional, respectively). On the other hand, the major compounds in the essential oil of the skin were limonene (15,893 and 16,953 mg $L^{-1}$, organic and conventional, respectively), β-myrcene (3060 and 2911 mg $L^{-1}$, organic and conventional, respectively) and γ-terpinene (2073 and 1797 mg $L^{-1}$, organic and conventional, respectively). These compounds have a marked aroma of grape, fruity, peach, lemon, orange, citrus and herbaceous [52] which gives a complex and attractive odor and aroma to this fruit. Of course, as expected, the concentration of the volatile compounds detected was much lower in the juice than in the essential oil.

In general, it can be observed how, both in the juice and in the essential oil, organically grown lemons have a higher concentration of key aromatic compounds, except for limonene in the essential oil, which was higher in the conventional samples.

The results show that the content of volatile compounds present, both in the juice and in the essential oil of the skin, is slightly influenced by agronomic practices during its cultivation.

Results obtained for limonene and γ-terpinene in the essential oil of the two lemon samples were similar to those reported by Di Vaio, Graziani, Gaspari, Scaglione, Nocerino and Ritieni [20] and Klimek-Szczykutowicz, Szopa and Ekiert [22], while our samples presented a higher concentration of β-myrcene. These differences may be due, among other factors, to the essential oil extraction methods used [53]. On the other hand, Aguilar-Hernández, Sánchez-Bravo, Hernández, Carbonell-Barrachina, Pastor-Pérez and Legua [18] obtained similar results for all the compounds detected in the essential oil in "Fino" and Verna lemon varieties.

**Table 5.** Concentrations (mg L$^{-1}$) of volatile compounds in lemon juice "Fino 49" cultivated under organic and conventional farming.

| Compound | RT | KI (Exp) | KI (Lit) | ANOVA | Conventional | Organic |
|---|---|---|---|---|---|---|
| 3-Heptanone | 3.717 | 891 | 890 | NS | 3.81 | 2.98 |
| α-Pinene | 4.570 | 943 | 939 | * | 8.48 b | 11.54 a |
| β-Pinene | 5.483 | 998 | 990 | * | 22.06 b | 27.18 a |
| β-Myrcene | 5.506 | 999 | 992 | *** | 32.25 b | 57.63 a |
| α-Phellandrene | 6.010 | 1019 | 1013 | NS | 1.07 | 1.19 |
| α-Terpinene | 6.244 | 1029 | 1019 | NS | 6.50 | 8.72 |
| p-Cymene | 6.467 | 1037 | 1026 | NS | 10.62 | 13.04 |
| Limonene | 6.559 | 1041 | 1033 | *** | 1086 b | 1310 a |
| trans-β-Ocimene | 6.848 | 1053 | 1050 | NS | 4.72 | 5.96 |
| γ-Terpinene | 7.253 | 1069 | 1062 | *** | 188.49 b | 252.85 a |
| 1-Octanol | 7.571 | 1081 | 1078 | NS | 6.80 | 5.54 |
| Terpinolene | 8.015 | 1099 | 1089 | *** | 36.07 b | 48.91 a |
| Linalool | 8.375 | 1110 | 1110 | * | 77.82 a | 35.18 b |
| Nonanal | 8.467 | 1112 | 1102 | ** | 13.05 b | 19.18 a |
| D-Fenchyl alcohol | 9.165 | 1133 | 1123 | NS | 8.62 | 13.41 |
| Citronellal | 10.077 | 1161 | 1165 | NS | 3.28 | 4.14 |
| 1-Nonanol | 10.722 | 1180 | 1173 | NS | 13.81 | 13.40 |
| Borneol | 10.967 | 1187 | 1177 | NS | 2.45 | 4.29 |
| Terpinen-4-ol | 11.172 | 1193 | 1184 | ** | 72.28 b | 84.48 a |
| α-Terpineol | 11.705 | 1208 | 1200 | * | 143.26 b | 198.45 a |
| Decanal | 11.868 | 1212 | 1207 | NS | 6.13 | 7.23 |
| Nerol | 12.723 | 1234 | 1228 | *** | 149.61 a | 62.06 b |
| Neral | 13.235 | 1247 | 1239 | NS | 5.39 | 3.41 |
| Geraniol | 13.686 | 1259 | 1255 | *** | 162.47 a | 59.99 b |
| Geranial | 14.375 | 1277 | 1277 | NS | 10.35 | 8.52 |
| Citronellyl acetate | 17.459 | 1353 | 1354 | NS | 3.71 | 2.83 |
| Neryl acetate | 17.858 | 1363 | 1366 | ** | 111.84 a | 82.15 b |
| Geranyl acetate | 18.666 | 1382 | 1381 | *** | 47.60 a | 20.39 b |
| trans-Caryophyllene | 20.507 | 1427 | 1418 | NS | 1.57 | 2.31 |
| trans-α-Bergamotene | 20.953 | 1438 | 1435 | *** | 2.68 b | 17.73 a |
| α-Farnesene | 23.708 | 1505 | 1509 | NS | 0.55 | 2.08 |
| β-Bisabolene | 23.993 | 1512 | 1509 | ** | 8.10 b | 37.36 a |
| Total | | | | ** | 2406 b | 2537 a |

NS: not significant at $p > 0.05$; *, ** and ***: significant at $p < 0.05$, 0.01 and 0.001, respectively. Values followed by different letters, within the same compound, were significantly different ($p < 0.05$). RT = retention time, KI (Exp.) = experimental Kovats index, KI (Lit.) = literature Kovats index.

**Table 6.** Concentrations (mg L$^{-1}$) of volatile compounds in lemon peel's oils "Fino 49" cultivated under organic and conventional farming.

| Compound | RT | KI (Exp) | KI (Lit) | ANOVA | Conventional | Organic |
|---|---|---|---|---|---|---|
| α-Thujene | 4.391 | 932 | 931 | NS | 92.05 | 104.78 |
| α-Pinene | 4.558 | 942 | 939 | NS | 438.94 | 465.35 |
| Camphene | 4.870 | 961 | 964 | NS | 15.77 | 17.08 |
| β-Pinene | 5.225 | 982 | 990 | NS | 525.32 | 563.78 |
| β-Myrcene | 5.388 | 992 | 992 | * | 2911 b | 3060 a |
| Octanal | 5.749 | 1009 | 1001 | NS | 37.38 | 37.79 |
| α-Phellandrene | 5.908 | 1015 | 1013 | NS | 10.67 | 12.09 |
| α-Terpinene | 6.141 | 1025 | 1019 | NS | 52.93 | 61.42 |
| p-Cymene | 6.344 | 1033 | 1026 | NS | 32.38 | 32.71 |
| Limonene | 6.526 | 1040 | 1033 | * | 16,953 a | 15,893 b |
| trans-β-Ocimene | 6.740 | 1048 | 1050 | NS | 25.85 | 30.44 |
| γ-Terpinene | 7.160 | 1065 | 1062 | * | 1797 b | 2073 a |
| 1-Octanol | 7.468 | 1077 | 1078 | NS | 9.43 | 7.93 |
| cis-Sabinene hydrate | 7.560 | 1081 | 1074 | NS | 18.01 | 23.00 |
| α-Terpinolene | 7.926 | 1095 | 1089 | ** | 89.60 b | 105.74 a |

**Table 6.** *Cont.*

| Compound | RT | KI (Exp) | KI (Lit) | ANOVA | Conventional | Organic |
|---|---|---|---|---|---|---|
| Linalool | 8.293 | 1107 | 1110 | ** | 133.20 b | 146.07 a |
| Nonanal | 8.485 | 1113 | 1102 | NS | 84.47 | 79.20 |
| Citronellal | 9.942 | 1157 | 1165 | * | 52.86 a | 35.59 b |
| Camphor | 10.033 | 1159 | 1143 | NS | 8.57 | 10.52 |
| 1-Nonanol | 10.274 | 1166 | 1173 | NS | 12.92 | 14.34 |
| Borneol | 10.928 | 1186 | 1177 | NS | 22.42 | 23.84 |
| Terpinen-4-ol | 11.141 | 1192 | 1184 | * | 67.10 b | 80.13 a |
| α-Terpineol | 11.681 | 1207 | 1200 | * | 210.76 b | 252.17 a |
| Decanal | 11.843 | 1211 | 1207 | * | 11.93 | 10.06 |
| Nerol | 12.714 | 1234 | 1228 | NS | 128.86 | 139.72 |
| Neral | 13.221 | 1247 | 1239 | ** | 754.04 b | 778.20 a |
| Geraniol | 13.689 | 1259 | 1255 | NS | 90.40 | 96.35 |
| Geranial | 14.378 | 1277 | 1277 | NS | 1024 | 1064 |
| Neryl acetate | 17.890 | 1363 | 1366 | NS | 109.52 | 119.32 |
| Geranyl acetate | 18.706 | 1383 | 1381 | NS | 42.73 | 42.03 |
| trans-Caryophyllene | 20.547 | 1428 | 1418 | NS | 26.29 | 32.26 |
| trans-α-Bergamotene | 20.988 | 1439 | 1435 | NS | 61.91 | 64.79 |
| α-Farnesene | 23.740 | 1506 | 1509 | NS | 6.41 | 12.63 |
| β-Bisabolene | 24.051 | 1514 | 1509 | NS | 84.91 | 93.36 |
| Total | | | | ** | 25,945 a | 25,585 b |

NS: not significant at $p > 0.05$; * and **: significant at $p < 0.05$ and 0.01, respectively. Values followed by different letters, within the same compound, were significantly different ($p < 0.05$). RT = retention time, KI (Exp.) = experimental Kovats index, KI (Lit.) = literature Kovats index.

### 3.4. Sensory Analysis

Of the total number of consumers (100), 44% were male and 56% were female, of which 39% were between 18 and 24 years old, 39% were between 25 and 35 years old and 22% were older than 36 years.

The color, bitterness and aftertaste of the samples did not have significant differences between the samples. On the other hand, consumers liked the lemon odor (6.9), lemon flavor (6.5) and sourness (5.5) more in the organically grown samples (Table 7). These data are correlated with those found in the results of volatile compounds, since the organic samples obtained the highest total concentration of these compounds. On the other hand, these results are not directly related to the values found in the analyses of the content of sugars and organic acids. Lemons from conventional cultivation had a higher concentration of sugars and organic acids than those from organic cultivation. However, consumers found the organically grown samples to be sourer. This may be due to the fact that, despite having a lower concentration of organic acids (the organic samples compared to the conventional ones), they also had a lower concentration of sugars, so it is possible that the total acid/sugar ratio is higher, and the acidity stands out. Likewise, consumers also preferred organic samples (6.2) for overall liking (global), the attribute that defines the final opinion of consumers about the overall quality of a sample. King, et al. [54] showed that small changes in sensory sourness can cause cognitive associations that lead consumers to relate this change with greater intensities of fruity and citrus flavors. Sourness is the sensory characteristic most appreciated by consumers in lemons, so it is justified that a higher perception of acidity is related to a higher degree of satisfaction.

When consumers were forced to choose which was their favorite sample, organic lemons were the most liked ones (78%) and, in the same way, organic lemons had 50% of consumers willing to pay, in contrast to only 33% presented by conventional lemons (Figure 1). Consumers mentioned that the main reasons for selecting the preferred sample (the most liked one) were: (i) lemon flavor (~67%), (ii) sourness (~44%) and (iii) lemon odor (~28%).

**Table 7.** Sensory affective test on conventional and organic lemon juice.

|              | Color | Lemon Odor | Lemon Flavor | Sourness | Bitterness | Aftertaste | Global |
|--------------|-------|------------|--------------|----------|------------|------------|--------|
| Conventional | 6.1   | 4.8 b      | 5.2 b        | 4.7 b    | 5.2        | 6.5        | 4.5 b  |
| Organic      | 6.4   | 6.9 a      | 6.5 a        | 5.5 a    | 4.8        | 6.4        | 6.2 a  |
| ANOVA        | NS    | ***        | *            | *        | NS         | NS         | **     |

NS: not significant at $p > 0.05$; *, ** and ***: significant at $p < 0.05$, 0.01 and 0.001, respectively. Values followed by different letters, within the same column, were significantly different ($p < 0.05$).

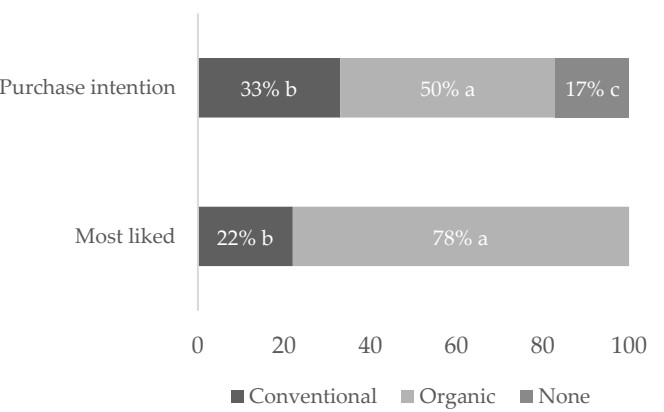
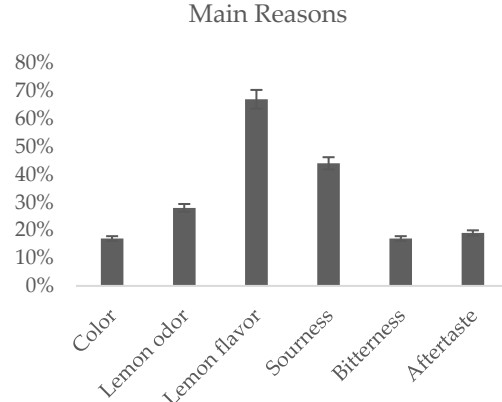

**Figure 1.** Consumer's preference, purchase intention and the main reasons behind their selection of organic and conventional lemons. Factors with different letters were significantly different ($p < 0.05$), Tukey's least significant difference test.

## 4. Conclusions

In general, the conventional lemon samples showed larger and elongated fruits, with a higher content of organic acids and sugars, while the organic fruits were rounder. There were no differences between the visual appearance of both fruits. On the other hand, lemons grown under organic conditions had a higher content of volatile compounds and greater acceptance by consumers. These results show that the agronomic practices carried out during lemon cultivation affect the functional and sensory quality of this fruit.

**Author Contributions:** Conceptualization, L.N.-A., J.M.-T., F.H. and E.S.; investigation, P.S.-B., L.N.-A., F.H. and E.S.; methodology, P.S.-B., J.M.-T., F.H. and E.S.; supervision, J.M.-T. and F.H.; writing—original draft, P.S.-B., L.N.-A., J.M.-T. and E.S.; writing—review & editing, E.S. All authors have read and agreed to the published version of the manuscript.

**Funding:** Paola Sánchez-Bravo was funded by the grant for the recall of the Spanish university system for the training of young doctors (Margarita Salas, 04912/2021) funded by the European Union-Next Generation EU, Ministry of Universities of Spain. The GCMS has been acquired thanks to Grant EQC2018-004170-P funded by MCIN/AEI/10.13039/501100011033 and by ERDF A way of making Europe.

**Institutional Review Board Statement:** The study was conducted according to the guidelines of the Declaration of Helsinki and approved by the Miguel Hernández University of Elche Ethics Committee in 2021. The ethical approval reference number for this study is PRL.DTA.ESN.03.20.

**Informed Consent Statement:** Informed consent was obtained from all subjects involved in the study. None of the data shared in this article can lead to the identification of any of the participants.

**Data Availability Statement:** Not applicable.

**Conflicts of Interest:** The authors declare no conflict of interest.

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
