# Peer review of "Effect of Organic and Conventional Production on the Quality of Lemon “Fino 49”"

_agronomy, doi:10.3390/agronomy12050980_

Round 1
Reviewer 1 Report
In the present paper, the research is interesting and scientific enough. Evaluation and detection of differences between organic and conventional farming is important due to political/social and economical expectations. The topic of the study fits to the area of “Agronomy”. The manuscript is well structured, written in good English, interesting to read and relevant. This paper provides novel information about the organic and conventional farming. However, the abstract is too general and should be renew about the relationships between chemical and sensory analysis. Introduction gives a short overview of the topic. Material and methods are well described, clear and easy to follow. Some clarification about the study site is needed. Results and discussion are clear, show most important significance of the research. Still, there is a need for more statistical analysis to study the impact of chemical compounds to sensory results. Tables and figure are suitable with minor modifications. The cited references should be more current (12 out of 43 are within the last 5 years). The main aim of the article has been achieved. My suggestion is that the manuscript does meet the requirements of Agronomy after major revision.
Abstract
The abstract is too general and should renew. There is a lack of relationships and conclusions between chemical compounds (organic acids, sugars, volatile compounds) and sensory tests results.
Introduction
Line 33-37. Authors describe two main varieties (“Fino” and “Verna”). Unfortunately, only the characteristics of the “Fino” are described below. It would be informative to read about another variety as well.
As the article largely contains the determination of different chemical compounds, there could be more information on the effects of the main chemical compounds on sensory tests.
Materials and Methods
Line 62-64. What are the geographical coordinates of these two orchards and what is the distance between them?
Line 66-68. Characteristics of “Fino 49” have been used in the comparison (e.g., larger, heavier, higher etc.), but remain confusing with variety. Is it “Verna” of “Fino”?
Line 75-88. The authors compare the water of two orchards in detail, but the soil characteristics are described for only one soil. In addition, pH of the soil should be clarified.
Line 89-111. Much different information. Used plant protection to control pests and diseases and amount of nutrients is hardly understandable. The table would help to understand it better.
Line 113. How many randomly selected trees are included in the study? Is it also fifty different trees?
Line 131-132. According to the text in manuscript, “Measurements were conducted in triplicate and values were expressed as mean ± standard error (SE).” Still, all means are shown without SE throughout the manuscript.
Results and Discussion
Table 1.-Table 6. Symbols “†” and “‡” is confusing. Different letters already indicate significant differences between treatments. In addition, data represents mean together with standard deviations (± SD) would be more informative to illustrate the amount of variability.
Line 197-198. Part of the results “Growing lemon in a conventional way has been shown to obtain fruits with greater weight (121.75 g)…” is remarkable lower than mention in section “Plant Material” where authors write “The fruits are larger and heavier (170 g) and have….”. What could be the reason for smaller fruits?
Line 217. The style of the reference (Galindo et al., 2015) does not meet the requirements of the “Agronomy”.
Results missing statistical analysis and conclusions between different compounds (organic acids, sugars, volatile compounds) and sensory analysis. For example, in the Introduction, authors refer that the flavor of lemon depends on the organic acids and sugars (line 47). Authors find out that fruits from conventional farming have higher content of organic acids and sugars. Results from sensory analysis showed lower scale of lemon flavor in conventional farming. Secondly, many volatile compounds have been qualified and quantified (Table 4 and Table 5) and detailed sensory test have been made (lemon odor, flavor, sourness, bitterness, aftertaste). According to data from this research, it is possible to find statistically the most important chemical compounds that play main effect on sensory results. In the end, lack of discussion about these relationships.
Results contain a large number of volatile compounds (Table 4 and Table 5). The recommendation is to use principal component analysis to visualize variation of volatile compounds under conventional and organic farming.
Line 320-321. Figure 1. Confusing is the sentence: “Factors with the same letter were not significantly different” because no factor with same letters is on the figure.
Author Response
The authors thank the reviewer for their helpful comments. All the suggestions have been considered and we are sure that the manuscript has been really improved.
Abstract
The abstract is too general and should renew. There is a lack of relationships and conclusions between chemical compounds (organic acids, sugars, volatile compounds) and sensory tests results.
Thank you very much for your thoughtful comment. The abstract has been modified.
Introduction
Line 33-37. Authors describe two main varieties (“Fino” and “Verna”). Unfortunately, only the characteristics of the “Fino” are described below. It would be informative to read about another variety as well.
Done as suggested. Please, see lines 49-50.
As the article largely contains the determination of different chemical compounds, there could be more information on the effects of the main chemical compounds on sensory tests.
The reviewer was right. The authors have included a paragraph in the introduction explaining the relationship between volatile compounds and sensory analysis. Please, see lines 65-69.
Materials and Methods
Line 62-64. What are the geographical coordinates of these two orchards and what is the distance between them?
Included as suggested. Please, see lines 91-93.
Line 66-68. Characteristics of “Fino 49” have been used in the comparison (e.g., larger, heavier, higher etc.), but remain confusing with variety. Is it “Verna” of “Fino”?
The reviewer was right. "Fino 49" is a clone of the "Fino" variety and is currently (last 10 years) the most cultivated in the Mediterranean area. In this article, the "Verna" variety has not been studied, only it is mentioned in the introduction since it is the one that represents the highest production. We have rewritten this part to avoid confusion. Please, see lines 79-80.
Line 75-88. The authors compare the water of two orchards in detail, but the soil characteristics are described for only one soil. In addition, pH of the soil should be clarified.
The reviewer is completely right. The authors apologize for this mistake. This mistake has been caused by a translation error, in which the values of soil from both farms were mixed. Now it is correct. Thank you for your comment. Please, see plant material section.
Line 89-111. Much different information. Used plant protection to control pests and diseases and amount of nutrients is hardly understandable. The table would help to understand it better.
Done as suggested. Please, see table 1.
Line 113. How many randomly selected trees are included in the study? Is it also fifty different trees?
The information has been included in the manuscript. Please, see the lines 145-151.
Line 131-132. According to the text in manuscript, “Measurements were conducted in triplicate and values were expressed as mean ± standard error (SE).” Still, all means are shown without SE throughout the manuscript.
Everything has been moved to the statistics section and is now all showed in statistics section. The ANOVA and the Tukey Test already showed the significant differences between the samples (taking into account the standard error). To simplify the results, the authors have preferred not to include standard deviations.
Results and Discussion
Table 1.-Table 6. Symbols “†” and “‡” is confusing. Different letters already indicate significant differences between treatments. In addition, data represents mean together with standard deviations (± SD) would be more informative to illustrate the amount of variability.
Symbols has been removed to avoid confusions.
Line 197-198. Part of the results “Growing lemon in a conventional way has been shown to obtain fruits with greater weight (121.75 g)…” is remarkable lower than mention in section “Plant Material” where authors write “The fruits are larger and heavier (170 g) and have….”. What could be the reason for smaller fruits?
For this experiment, the medium fruits of the tree were taken, so that they were representative of the entire harvest. The value of the weight given in the introduction is merely informative to be able to compare the varieties, according to the bibliography.
Line 217. The style of the reference (Galindo et al., 2015) does not meet the requirements of the “Agronomy”.
The reference has been changed to MDPI style. Now it is correct. Please, see line 261.
Results missing statistical analysis and conclusions between different compounds (organic acids, sugars, volatile compounds) and sensory analysis. For example, in the Introduction, authors refer that the flavor of lemon depends on the organic acids and sugars (line 47). Authors find out that fruits from conventional farming have higher content of organic acids and sugars. Results from sensory analysis showed lower scale of lemon flavor in conventional farming. Secondly, many volatile compounds have been qualified and quantified (Table 4 and Table 5) and detailed sensory test have been made (lemon odor, flavor, sourness, bitterness, aftertaste). According to data from this research, it is possible to find statistically the most important chemical compounds that play main effect on sensory results. In the end, lack of discussion about these relationships.
This part has been expanded by the authors to better explain the relationship of the results obtained. Please see the “sensory analysis” section.
Results contain a large number of volatile compounds (Table 4 and Table 5). The recommendation is to use principal component analysis to visualize variation of volatile compounds under conventional and organic farming.
The authors do not know how to perform a PCA with only two components. This type of analysis is used to simplify the data when there are a large number of components, but when there are only two components, the software that we use does not allow it to be done.
Line 320-321. Figure 1. Confusing is the sentence: “Factors with the same letter were not significantly different” because no factor with same letters is on the figure.
The sentence has been modified. Please, see Figure 1 footnote.
Reviewer 2 Report
The text describes a comparison of conventional and organic lemons produced in Spain. The comparison is based on four components: morphology, function, aroma and sensory properties.
As organic agriculture is on the raise – see, for instance, the ambitious targets mentioned in the EU Farm to fork strategy – it is important to get a firm grasp on the differences between the production processes but also between the resulting food products. Thus the current study is relevant and topical.
Overall, the text is well-written and clearly structured. Still, some issues are worth noting.
- Abstract: mention the geographical focus of the study (Spain)
- Line 23: ‘on a global scale’ instead of ‘…within the world population’
- Line 27: ‘smaller volumes of food’ instead of ‘less amount of food’
- Line 29: ‘among other things’ instead of ‘among others’
- Line 59 Section 2:
- please also mention when the data were collected – which year?
- It may be interesting to describe average weather conditions (rain, sun, temperature…)
- Line 103-111: why are fertilizers fully capitalized?
- Line 113: why 50? Did you do power calculations?
- Line 177: I assume the consumer test was blind but this is not mentioned specifically. Was the order the same for all consumers or was it randomized? Did all consumers test two sample juices? Or more than two?
- Line 178: the gender distribution described here is different from that described on line 296
- Line 179: the age distribution described here is different from that described on line 298-299
- Line 184: some characteristics are desirable (lemon flavor) while other characteristics are not (e.g. bitterness). How is this taken into account?
- Line 185: ‘purchase intention’ is not the same as ‘willingness to pay’
- Line 185: were the questions about purchase intention and global satisfaction still blind?
- Line 192: what do you mean by ‘the analysis was run in triplicate’? Wouldn’t a statistical test give exactly the same outcome even if you performed it three times?
- Line 306: which ‘three samples’ are referred to? Conventional vs organic vs ?
- Figure 1: Replace ‘willingness to pay’ by ‘purchase intention’
Author Response
The authors thank the reviewer for their helpful comments. All the suggestions have been considered and we are sure that the manuscript has been really improved.
- Abstract: mention the geographical focus of the study (Spain)
Added as suggested.
- Line 23: ‘on a global scale’ instead of ‘…within the world population’
Done as suggested. Please, see line 28.
- Line 27: ‘smaller volumes of food’ instead of ‘less amount of food’
Done as suggested. Please, see line 32.
- Line 29: ‘among other things’ instead of ‘among others’
Done as suggested. Please, see line 34.
- Line 59 Section 2:
- please also mention when the data were collected – which year?
Include as suggested. Please, see line 145.
- It may be interesting to describe average weather conditions (rain, sun, temperature…)
Include as suggested. Please, see lines 94-96.
- Line 103-111: why are fertilizers fully capitalized?
Fertilizers have been written without capital letters. Please, see lines 123-143.
- Line 113: why 50? Did you do power calculations?
This sampling is based on previous experiences with this fruit and with other types of citrus, as well as the existing bibliography.
- Line 177: I assume the consumer test was blind but this is not mentioned specifically. Was the order the same for all consumers or was it randomized? Did all consumers test two sample juices? Or more than two?
All consumers tasted both samples in random order. The samples were coded in a 3-digit number and served in odor-free glasses.
- Line 178: the gender distribution described here is different from that described on line 296
The reviewer was right. It was a mistake, the real data are those that appear in the results and discussion section of sensory analysis (section 3.4).
- Line 179: the age distribution described here is different from that described on line 298-299
The reviewer was right. It was a mistake, the real data are those that appear in the results and discussion section of sensory analysis (section 3.4).
- Line 184: some characteristics are desirable (lemon flavor) while other characteristics are not (e.g. bitterness). How is this taken into account?
We are very sorry. The authors do not understand what the reviewer means by this phrase. In that line or in any other do we refer to unwanted attributes.
- Line 185: ‘purchase intention’ is not the same as ‘willingness to pay’
The authors want to express "purchase intention".
- Line 185: were the questions about purchase intention and global satisfaction still blind?
Yes. The complete sensory study was carried out with the samples coded with 3-digit codes.
- Line 192: what do you mean by ‘the analysis was run in triplicate’? Wouldn’t a statistical test give exactly the same outcome even if you performed it three times?
The analyzes carried out during the study were performed in triplicate, not the ANOVA. It was a translation error. Sorry, the sentence has been corrected. Please, see line 235.
- Line 306: which ‘three samples’ are referred to? Conventional vs organic vs ?
The reviewer is right, it was a mistake. There were only two samples. Please, see line 367.
- Figure 1: Replace ‘willingness to pay’ by ‘purchase intention’
Changed as suggested. Please, see figure 1.
Reviewer 3 Report
Dear Authors, thank you for the opportunity to contribute to the development of your work. I think, that the topic itself is very interesting, the experiment is well designed, but the interpretation of the results and especially the discussion slightly lost its focus and does not deal with the comparison of the two production systems.
General comments:
The title looks a bit strange with the variety name in brackets. Please remove it, or add it with inverted commas to the title.
I find the introduction section very short. It does not deal with the main topic of the study, rather gives a general overview about the importance of lemon. Please focus on the organic vs. conventional issue.
The materials and methods reqires further clarifications. Regarding the two farms, it is not clear, how far these are from each other, i.e. how similar the soil and microclimatic parameters are. It is impossible, that the soil of both farms is identical in terms of SOM, nitrate content, C/N ration and so on, see lines 75-79. No information is provided about the age of orchards. Also, the date of sampling is not given. It is very misleading, that farm names are used throughout the whole text for the identification of production systems. I would suggest to use abbreviations instead, which clearly shows, which production system are you talking about. The plant protection measures, as well as the fertilization should be put into a table to enhance the readability and comparability of the systems. I think, the names of fertilizers are unneccesary to write with capital letters. Regarding the sensory analysis it is unclear how many samples were assessed. Later you wrote three samples, but you had only two treatments. Please clarify.
Table 4 and 5: I would prefer to rank the compounds according to their quantity (either organic or conventional) to show their importance for the first look.
A great weakness of the study, that it deals extensively with the comparison of the compound levels with those of the previous studies, but does not deal with the comparison of their own outcomes with those of other organic vs. conventional studies.
Author Response
The authors thank the reviewer for their helpful comments. All the suggestions have been considered and we are sure that the manuscript has been really improved.
General comments:
The title looks a bit strange with the variety name in brackets. Please remove it, or add it with inverted commas to the title.
Done as suggested.
I find the introduction section very short. It does not deal with the main topic of the study, rather gives a general overview about the importance of lemon. Please focus on the organic vs. conventional issue.
Thanks for your comment. The introduction has been enlarged.
The materials and methods reqires further clarifications. Regarding the two farms, it is not clear, how far these are from each other, i.e. how similar the soil and microclimatic parameters are. It is impossible, that the soil of both farms is identical in terms of SOM, nitrate content, C/N ration and so on, see lines 75-79. No information is provided about the age of orchards. Also, the date of sampling is not given. It is very misleading, that farm names are used throughout the whole text for the identification of production systems. I would suggest to use abbreviations instead, which clearly shows, which production system are you talking about. The plant protection measures, as well as the fertilization should be put into a table to enhance the readability and comparability of the systems. I think, the names of fertilizers are unneccesary to write with capital letters. Regarding the sensory analysis it is unclear how many samples were assessed. Later you wrote three samples, but you had only two treatments. Please clarify.
The reviewer is completely correct. The authors apologize for the error. This failure has been caused by a translation error, in which the values of both farms were mixed. Now it is correct. The names of fertilizers have been wrote in normal letters. The reviewer is right, it was a mistake. There were only two samples. Please, see materials and method section.
Table 4 and 5: I would prefer to rank the compounds according to their quantity (either organic or conventional) to show their importance for the first look.
Although we greatly appreciate your comment, the authors prefer to express the data of volatile compounds based on their retention time, because in this way they are also ordered based on their retention index and it is easier to compare the results.
A great weakness of the study, that it deals extensively with the comparison of the compound levels with those of the previous studies, but does not deal with the comparison of their own outcomes with those of other organic vs. conventional studies.
We have expanded the information as you have rightly suggested. Please, see lines 266-270, 285-287, 298-300.
Round 2
Reviewer 1 Report
The manuscript has been improved and is ready to accept for publication.
Author Response
Thank you vrey much to help us to improve this manuscript.
Reviewer 2 Report
The revisions executed by the authors have clarified several issues and it is now clear how the study was implemented and how the results can be interpretated.
Still I have noted some minor issues:
Line 37 – who is referred to by ‘they’ in this sentence?
Line 49-50 – the text that is added here starting with ‘while Verna variety…’ does not fit logically with the first part of this sentence. I would recommend making it a separate sentence.
Line 235 -236: I still don’t understand which part of the analysis was performed in triplicate. What do you mean with ‘all of the determinations’ in the context of a survey? As it was not the statistical test, can I assume that you asked each respondent to assess the sensory characteristics of the sample juice three times? So each respondent assessed the same juice three times? Please clarify.
Line 371-372: what do you mean by ‘organic lemons had 50% consumers’ willingness to pay’? Please check the grammar of this sentence and the replace ‘willingness to pay’ by ‘willingness to buy’ or ‘positive intentions to purchase’ or ‘openness to buy’.
Author Response
The revisions executed by the authors have clarified several issues and it is now clear how the study was implemented and how the results can be interpretated. Still I have noted some minor issues:
Line 37 – who is referred to by ‘they’ in this sentence?
Sorry for the mistake. The correct sentence is: Currently, there have increased the number of organic farms and therefore the extent of farmland. Sentence has been modified. Please, see line 37.
Line 49-50 – the text that is added here starting with ‘while Verna variety…’ does not fit logically with the first part of this sentence. I would recommend making it a separate sentence.
Done as suggested. Please, see lines 49-50.
Line 235 -236: I still don’t understand which part of the analysis was performed in triplicate. What do you mean with ‘all of the determinations’ in the context of a survey? As it was not the statistical test, can I assume that you asked each respondent to assess the sensory characteristics of the sample juice three times? So each respondent assessed the same juice three times? Please clarify.
Reviewer is right. The determinations that were made in triplicate were those corresponding to the physical-chemical analyses, organic acids, sugars and volatile compounds. This is indicated in these sections.
In the sensory case, each judge tested each of the samples only once. That is, every panelist tested 2 samples (organic and conventional). That is why 100 judges were needed, in order to have a repeatability according to the existing deviation due to the variability of responses in this type of test.
To avoid reader confusion, thanks to your valuable comment, we have removed this sentence from the statistical analysis section and have kept the specific ones for each of the analyses.
Line 371-372: what do you mean by ‘organic lemons had 50% consumers’ willingness to pay’? Please check the grammar of this sentence and the replace ‘willingness to pay’ by ‘willingness to buy’ or ‘positive intentions to purchase’ or ‘openness to buy’.
Yes, reviewer is right. The sentence has been changed to improve their grammar. Now, the sentence is: When consumers were forced to choose which was their favorite sample organic lemons were the most liked ones (78 %). Besides, 50 % of consumers were willingness to buy organic lemons, while only 33 % of them showed an increase purchase intention on conventional ones.
Please, see the changes in lines 370-373.
We take advantage of these lines to thank you for your dedication when reviewing this article. Thanks to your valuable comments, the article has been greatly improved.
Thank you very much.
Reviewer 3 Report
Dear Authors, thank you for considering my suggestions. I think, that the ms has been developed sufficiently. Only one small remark, that please put 'variety names' between these single superscript commas instead of inverted double commas. I might have wrongly advised it previously.
Author Response
Dear Authors, thank you for considering my suggestions. I think, that the ms has been developed sufficiently. Only one small remark, that please put 'variety names' between these single superscript commas instead of inverted double commas. I might have wrongly advised it previously.
Done as suggested. Thank you so much for your time with our manuscript. Thank you very much to help us to improve this manuscript.